# Pain burden, sensory profile and inflammatory cytokines of dogs with naturally-occurring neuropathic pain treated with gabapentin alone or with meloxicam

**Hélène L. M. Ruel[1], Ryota Watanabe[1], Marina C. Evangelista[1], Guy Beauchamp[2], Jean-Philippe Auger[3], Mariela Segura[3], Paulo V. Steagall[1] ***

1 Department of Clinical Sciences, Faculté de médecine vétérinaire, Université de Montréal, Saint-Hyacinthe, Québec, Canada, 2 Faculté de médecine vétérinaire, Université de Montréal, Saint-Hyacinthe, Québec, Canada, 3 Research Group on Infectious Diseases in Production Animals (GREMIP) and Swine and Poultry Infectious Diseases Research Centre (CRIPA), Faculté de médecine vétérinaire, Université de Montréal, Saint-Hyacinthe, Québec, Canada

* paulo.steagall@umontreal.ca

**Data Availability Statement:** All relevant data are within the manuscript and its Supporting Information files.

## Abstract

Canine neuropathic pain (NeuP) has been poorly investigated. This study aimed to evaluate the pain burden, sensory profile and inflammatory cytokines in dogs with naturally-occurring NeuP. Twenty-nine client-owned dogs with NeuP were included in a prospective, partially masked, randomized crossover clinical trial, and treated with gabapentin/placebo/gabapentin-meloxicam or gabapentin-meloxicam/placebo/gabapentin (each treatment block of 7 days; total 21 days). Pain scores, mechanical (MNT) and electrical (ENT) nociceptive thresholds and descending noxious inhibitory controls (DNIC) were assessed at baseline, days 7, 14, and 21. DNIC was evaluated using ΔMNT (after-before conditioning stimulus). Positive or negative ΔMNT corresponded to inhibitory or facilitatory pain profiles, respectively. Pain scores were recorded using the Client Specific Outcome Measures (CSOM), Canine Brief Pain Inventory (CBPI), and short-form Glasgow Composite Measure Pain Scale (CMPS-SF). Data from baseline were compared to those of sixteen healthy controls. ΔMNT, but not MNT and ENT, was significantly larger in controls (2.3 ± 0.9 N) than in NeuP (-0.2 ± 0.7 N). The percentage of dogs with facilitatory sensory profile was similar at baseline and after placebo (61.5–63%), and between controls and after gabapentin (33.3–34.6%). The CBPI scores were significantly different between gabapentin (CBPI $_{pain}$ and CBPI $_{overall\ impression}$) and/or gabapentin-meloxicam (CBPI $_{pain}$ and $_{interference}$) when compared with baseline, but not placebo. The CBPI scores were not significantly different between placebo and baseline. The concentration of cytokines was not different between groups or treatments. Dogs with NeuP have deficient inhibitory pain mechanisms. Pain burden was reduced after gabapentin and/or gabapentin-meloxicam when compared with baseline using CBPI and CMPS-SF scores. However, these scores were not superior than placebo, nor placebo was superior to baseline evaluations. A caregiver placebo effect may have biased the results.

**Funding:** This study was supported by Boehringer Ingelheim (Canada) Ltd and MITACS through the Mitacs Accelerate Program, and the American Kennel Club Canine Health Foundation (#CHF : 02353-A) (Paulo V Steagall, Hélène LM Ruel). The funders had no role in data collection and analysis, decision to publish, or preparation of the manuscript.

**Competing interests:** This study was funded by Boehringer Ingelheim (Canada) Ltd and MITACS through the Mitacs Accelerate Program, and the American Kennel Club Canine Health Foundation. This does not alter our adherence to PLOS ONE policies on sharing data and materials.

# Introduction

Neuropathic pain (NeuP) is caused by a lesion or disease of the somatosensory system [1]. Its diagnosis relies on sensory examination of nerve fibers involved in nociception/proprioception for both loss (i.e. hypoesthesia and hypoalgesia) and gain of function (i.e. hyperalgesia and allodynia) via quantitative sensory testing (QST) [2]. In brief, QST is a psychophysical method that evaluates the somatosensory function from receptor to cortex using calibrated innocuous or noxious stimuli. It offers useful insight into the underlying pain mechanisms and the characterization of painful conditions [3]. For example, it is possible to stratify human patients with peripheral NeuP by categories of phenotypes using cluster analysis of their mechanical and thermal sensory profiles instead of a disease etiology-based classification [4]. Therefore, response to therapy can be predicted in precision or personalized medicine based on the specific patient sensory profile [5]. Additionally, changes in QST before and after the application of a conditioning stimulus provide useful information about the diffuse noxious inhibitory control (DNIC) as a representation of central descending modulatory pain mechanisms. The latter could predict people's response to drugs acting on central pain modulation [6]. It has been proposed that inflammatory cytokines play a role in the development and maintenance of NeuP and could be an avenue for future therapeutic options [7].

The diagnosis of NeuP in veterinary and cognitively-impaired human patients is a challenge. In companion animal medicine, the disease is diagnosed after appropriate physical, neurological and magnetic resonance imaging (MRI) examination, and clinical signs of pain and allodynia [8]. In dogs, NeuP can be caused by spinal cord disease, chronic musculoskeletal conditions and peripheral neuropathies, among others. Treatment recommendations for this disease in companion animals are mostly based on case-series, review articles, anecdotal reports and scientific evidence from humans. Gabapentinoids (e.g. gabapentin) and tricyclic antidepressants (e.g. amitriptyline) have been suggested as the first line of treatment of this disease [8]. Non-steroidal (NSAIDs) or steroidal anti-inflammatory drugs and antagonists of N-methyl-D-aspartate receptors (e.g. amantadine) have been also recommended [8]. Thus, a combination of a NSAID (e.g. meloxicam) and gabapentin are often anecdotally used in the treatment of NeuP conditions that are refractory to therapy with gabapentin alone. However, the efficacy of these treatments for NeuP has not been systematically studied in veterinary medicine.

The aims of this study were to evaluate the pain burden, sensory profile and inflammatory cytokines of dogs with NeuP before and after treatment with placebo, gabapentin alone or gabapentin-meloxicam. The sensory (QST) and inflammatory profiles of dogs with NeuP at presentation were compared with a population of healthy controls. Pain burden was determined using clinical pain assessment tools (pet owner and veterinary assessments). The hypotheses were that NeuP presents different sensory profile (i.e. hypo- or hyperalgesia) when compared with healthy controls and that treatment with gabapentin alone or with meloxicam alters this profile. Pain scores are expected to improve after treatment with gabapentin or gabapentin-meloxicam when compared with baseline (initial presentation) and placebo using both owner and veterinary assessments. Pro-and anti-inflammatory cytokine concentrations would be higher and lower, respectively, in dogs with NeuP than in controls. The serum concentrations of gabapentin were measured as an indirect method to assess treatment compliance.

# Methods

## Ethical statement

This study was approved by the local animal care committee of the Faculty of Veterinary Medicine, Université de Montréal (16-Rech-1835 and 16-Rech-1848) and was conducted between October

2016 and July 2018. The study is reported according to the CONSORT guidelines for randomized, clinical trials [9]. This was a prospective, partially masked, randomized crossover clinical trial.

## Animals

Thirty-two client-owned dogs were admitted to the veterinary teaching hospital (Centre Hospitalier Universitaire Vétérinaire) of the Université de Montréal. Dogs were recruited after physical and neurological examinations by a board-certified veterinary neurologist (H.L.M. R.). Owner's written consent was obtained for each patient.

Sixteen client-owned healthy control dogs (4.8 ± 2.1 years; 32 ± 16.7 kg; six males and ten females) were recruited simultaneously and their data were used for comparison. They were considered healthy based on history, physical, orthopedic and neurological examinations and did not received any analgesic treatment at least 30 days prior to recruitment. Exclusion criteria were the same as those described below for dogs with NeuP. Data for these individuals were previously reported as part of the validation of our methodology [10].

## Inclusion and exclusion criteria

Inclusion criteria were based on specific body weight ($\geq 4$ kg), age ($> 6$ months) and the owner's option for medical management of NeuP. Dogs were included if the duration of painful clinical signs was $\geq 4$ weeks and if a neurological lesion was found in the MRI consistent with the previous neurological examination and clinical signs of pain. Exclusion criteria included pregnancy, lactation, aggressive behavior, anxiety, history of pacemaker placement, systemic disease including chronic renal and hepatic disease, suspected immune-mediated disorders or any clinically relevant comorbidity, and significant changes in hematology and serum biochemistry analysis. Patients receiving treatments were weaned off medications at least 7 days (steroidal anti-inflammatory drugs), 24 hours (gabapentin), 72 hours (NSAIDs) and at least 60 minutes (remifentanil) before the clinical trial had begun.

## Treatments

Each dog was randomly allocated to treatment groups 1 or 2 (Table 1). Randomization was performed using balanced permutations (www.randomization.com). Each treatment was divided into three blocks of 7 days to include gabapentin or gabapentin-meloxicam (either first or third block) or placebo (always during the second block allowing a "wash-out" period between the first and third blocks). The total duration of the study was 21 days. Resting was recommended for all dogs (Fig 1).

Treatments were placed in pill dispensers and given to owners one week at a time. The capsules of 50, 100, 300 mg and tablets of 600 mg of gabapentin, and tablets of 1 and 2.5 mg of meloxicam were used. Drugs were administered orally (PO) at a targeted dose of 10 mg/kg every 8 hours for gabapentin (Gabapentin, Apotex®, Canada) and 0.2 mg/kg once followed

**Table 1. Treatment groups of a prospective, randomized, partially masked, placebo-controlled clinical trial in dogs with naturally-occurring presumptive neuropathic pain.**

|  | 1st block | 2nd block | 3rd block |
|---|---|---|---|
| Treatment group 1 | gabapentin (10 mg/kg every 8h, PO) + placebo tablets (every 24h, PO) | placebo capsules (every 8h, PO) + placebo tablets (every 24h, PO) | gabapentin (10 mg/kg every 8h, PO) + meloxicam (0.2 mg/kg PO followed by 0.1 mg/kg every 24h, PO) |
| Treatment group 2 | gabapentin (10 mg/kg every 8h, PO) + meloxicam (0.2 mg/kg PO followed by 0.1 mg/kg every 24h, PO) | placebo capsules (every 8h, PO) + placebo tablets (every 24h, PO) | gabapentin (10 mg/kg every 8h, PO) + placebo tablets (every 24h, PO) |

Oral administration (PO).

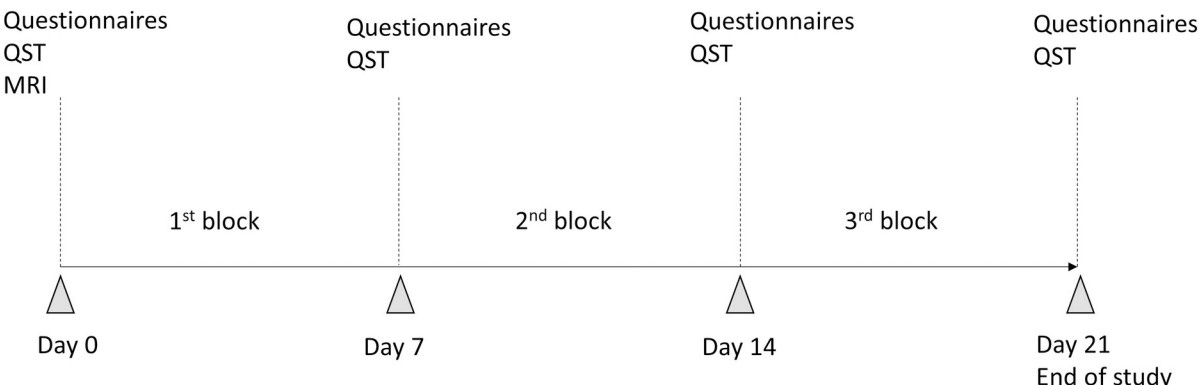

**Fig 1. Timeline of the study.** Dogs were randomized to receive either treatment 1 or 2. Pain assessment and Quantitative Sensory Testing (QST) were evaluated after each block of treatment (7 days). Abbreviations: QST, quantitative sensory testing (including mechanical and electrical nociceptive thresholds and assessment of the descending noxious inhibitory controls); MRI, magnetic resonance imaging.

by 0.1 mg/kg every 24 hours for meloxicam (Metacam, Boehringer Ingelheim Inc) (nearest whole capsule or up to one fourth of a tablet). Placebo compounds of dextrose were administered in tablets and/or capsules so that owners were masked to the treatment. The board-certified veterinary neurologist who participated in the study design was masked to the first and third (active treatments), but not to the second block (placebo).

## Quantitative sensory testing (QST)

QST was performed after physical and neurological examination and before the MRI at initial presentation (baseline, day 0) and following each treatment block (days 7, 14 and 21) (Fig 1).

Dogs were acclimated to the testing room for 10 minutes before the experimentation and had free access to water. For QST, they were positioned either in semi-sternal position or in lateral recumbency over a mat [10]. Nociceptive stimulations were stopped as soon as behavioral changes in response to stimuli were observed (looking at the probe, voluntary movement away from the probe, attempts to bite, etc.) [10].

The feasibility, intra- and inter-observer reliability, test-retest and sham-testing of our QST methodology have been previously reported [10]. Stimulation was applied to the dorsal aspect of the metacarpus and the plantar aspect of the metatarsus above the plantar pad bilaterally after clipping. The order of QST modality (electrical nociceptive thresholds, ENT; mechanical nociceptive thresholds, MNT), the limb and the side (right/left) of stimulation were randomized according to a random permutation generator (www.randomization.com). The observer graded each response to QST as poor (score 0), fair (score 1) or good (score 2) [10]. Replicates were obtained 60 seconds apart. If one of the responses received a score of 0 or 1, a third measurement was obtained 60 seconds later. Results with score 0 were not considered for statistical analysis. Outcome data for MNT and ENT were the mean of all measurements from all limbs, obtained with a score ≥1.

*Electrical nociceptive thresholds* — The stimulation was provided using a transcutaneous electrical nerve stimulator (TENS unit; Intelect® Vet two channel combo unit, Chattanooga, Guildford, Surrey, UK) in the VMS™ mode (View, Tempe, AZ, USA). The stimulation was delivered via two adhesive electrodes and consisted in a symmetrical biphasic waveform with a 100 μsec interphase. Settings were adjusted to a CC mode using a frequency of 200 Hz, phase duration of 20 μsec and a ramp of 0 seconds. The current was increased gradually until a behavioral response was observed, or until the cut-off of 150 mA was reached after 2 minutes.

*Mechanical nociceptive threshold (MNT) and diffuse noxious inhibitory controls (DNIC)* — For MNT, increasing pressure was applied perpendicular to the skin with an algometer (Bioseb, Vitrolles, France) with a flat tip of 3.5 mm diameter until a behavioral response was observed or the cut-off of 20 N reached.

The assessment of DNIC was based on the difference in MNT applied to one of the thoracic limbs before and after a conditioning stimulus. The conditioning stimulus was performed by placing an adult blood pressure cuff around the humerus and inflated it up to 200 mmHg for 60 seconds using a sphygmomanometer. After 3 minutes, the MNT was repeated on the same limb. The ΔMNT (after–before conditioning stimulus) was used as an outcome for the assessment of DNIC. When MNT was not obtained either pre- or post-conditioning stimulus for a dog, ΔMNT was not recorded. The percentage of positive and negative ΔMNT was calculated for each group. The DNIC was applied to the "least affected thoracic limb". The latter was based on neurological examination and localization of the lesion on the MRI. Increases in MNT after the conditioning stimulus are expected in healthy individuals with functional DNIC (i.e. functional inhibitory conditioned pain modulation), based on the "pain-inhibits-pain" paradigm [11].

The board-certified veterinary neurologist had previous training in QST in dogs [10]. This individual was responsible for identifying behavioral changes associated with nociceptive stimulation. This observer was not aware of stimuli intensity during testing. Two other individuals (M.C.E., R.W.) were involved in the QST: one was responsible for mild restraint of dogs during testing whereas the other controlled the electrical stimulation as previously reported [10]. They were also both responsible for randomization, recording nociceptive thresholds, preparation of the pill dispensers and compilation of results.

## Pain assessment tools (questionnaires)

At each visit (days 0, 7, 14 and 21), dog owners were asked to complete the client specific-outcome measures (CSOM) [12] and the French version of the Canine Brief Pain Inventory (CBPI) [13,14]. To complete the CSOM, owners listed three activities that were impaired due to pain or that elicited pain (e.g. getting up from lying down, jumping into the owner's car). The degree of difficulty to perform each activity (no problem, mildly problematic, moderately problematic, severely problematic or impossible) was followed weekly. The CBPI assesses pain severity, interference of pain on function (locomotion) and the owner's global impression about the dog's quality of life ("overall impression"). For "interference", questions regarding the dog's ability to run and to climb stairs were excluded since resting was recommended during the study. Therefore, the sections "pain" (CBPI $_{pain}$) and "interference" (CBPI $_{interference}$) contained each four questions scored on a 10-point scale (higher scores corresponding to greater difficulties/pain). The "overall impression" (CBPI $_{overall\ impression}$) was graded as poor, fair, good, very good and excellent. Additionally, the short-form Glasgow Composite Measure Pain Scale [CMPS-SF] [15] was completed at each visit by the veterinarian.

During the study, inadequate analgesia could be reported by the owners if they felt that clinical signs of pain persisted and were similar to presentation. In that case, a re-evaluation was scheduled at the earliest convenience and physical/neurological examination, pain scoring and QST repeated. If analgesic failure was observed with gabapentin-meloxicam during the first block, the dog was excluded from the trial. If it happened during the second block (placebo), the third block would start immediately. If it occurred during the third block, the study was finalized, and the dog treated according to the clinician's discretion. If owners reported pain during the withdrawal period (before entering the study), dogs were hospitalized to receive an intravenous infusion (CRI) of remifentanil as needed to alleviate pain until the study could be

started. Initial assessment would then be performed at least 60 minutes after the cessation of the administration of remifentanil. The choice of this drug as rescue analgesia was based on recent evidence that remifentanil was not associated with opioid-induced hyperalgesia in dogs and the convenience of its short half-life, allowing testing shortly after the cessation of the CRI and thus, minimizing the period without treatment of pain for the patient [16].

## Serum concentrations of gabapentin and inflammatory cytokines

Blood was collected by venipuncture into a sterile 3 mL anticoagulant-free glass tube (Mono-ject Blood Collection Tube; Covidien Canada, Saint-Laurent, QC, Canada) at each visit (day 0, 7, 14 and 21). Samples were allowed to clot at room temperature for at least 30 minutes before being centrifuged at 3000 rpm for 10 minutes. Subsequently, serum was aliquoted and stored at -70˚C in cryovials. Gabapentin was extracted from dog serum using a protein precipitation technique, separated by chromatography and then identified by mass spectrometry. (S1 File).

Serum samples were analyzed for concentrations of GM-CSF, IFN-γ, IL-2, IL-6, IL-7, IL-8, IL-15, IP-10, KC-like, IL-10, IL-18, MCP-1, and TNF-α using a pre-mixed Milliplex 13-plex Canine Magnetic Bead Panel (Millipore, Burlington, USA) according to the manufacturer's instructions. Acquisition was performed on the MAGPIX platform (Luminex®) and data analyzed using the MILLIPLEX Analyst 5.1 software (Upstate Group/Millipore). Standard curves and quality control checking were performed. Analytes with more than 50% out of range concentrations were excluded from statistical analyses. Cytokines of dogs with visible inflammatory conditions (severe oral inflammatory disease, dermatological problems such as skin allergies and otitis) were excluded from the statistical analysis.

## Statistical analysis

Data were analyzed using SAS (version 9.3; SAS Institute, Cary, NC, USA). A mixed linear model was used to analyze ENT, MNT and ΔMNT with treatment as the main effect and sex, age and body weight as covariates and dog ID as random effect. A mixed linear model was also used to assess the effects of treatment order with treatments and treatment order as main effects and age, sex and body weight as covariates. Additionally, a linear model was used to compare ENT, MNT and ΔMNT between healthy controls and NeuP using age, sex and body weight as covariates. The level of statistical significance was set at 5%. Incomplete questionnaires for pain assessment were excluded from the analysis. For the CSOM, responses were converted into a numerical scale ranging from 1 to 5, as previously described [12], with 1 = no problem, 2 = mildly problematic, 3 = moderately problematic, 4 = severely problematic, and 5 = impossible. The total CSOM score represented the sum of scores for each of the three activities.

Each section of the CBPI (namely CBPI $_{pain}$, CBPI $_{interference}$ and CBPI $_{overall\ impression}$) was analyzed separately. Grades for CBPI $_{overall\ impression}$ (poor, fair, good, very good and excellent) were translated to rank scores from 1 to 5 (poor: 1 to excellent: 5). Data for CBPI $_{overall\ impression}$ were analyzed with the Mantel-Haenszel chi-square followed pairwise comparisons using the sequential Benjamini-Hochberg procedure to adjust alpha levels. Data from CSOM, CBPI $_{pain}$ and CBPI $_{interference}$ and CMPS-SF were analyzed using a mixed linear model with treatment as the main effect and age, sex and body weight as covariates followed by Tukey's post-hoc tests when appropriate.

Data for serum concentrations of inflammatory cytokines were normalized using $\log_{10}$ transformation and compared afterwards. When measures obtained were out of range, they were replaced by the lowest value extrapolated by the software minus 0.01 in order to avoid missing data (and inherent bias). Cytokine analyses were performed using nonparametric test

when the distribution of data remained asymmetrical after $\log_{10}$ transformation (TNF-α). Otherwise, linear models were used (GM-CSF, IFN-γ, IL-2, IL-6, IL-7, IL-8, IL-15, IP-10, KC-like, IL-10, IL-18, MCP-1). Comparisons between treatments were performed using mixed linear models for all analytes, except for TNF-α, where Friedman test was used. The association between concentrations of cytokines and pain scores was assessed with Spearman correlation for CMPS-SF and CBPI $_{overall\ impression}$ which displayed a non-normal distribution and represented ordinal data. Furthermore, considering the absence of treatment effect on cytokine levels, data from NeuP and controls were pooled together to increase the sample size and avoid repeated measures for these parameters. Mixed linear models were used to analyze the association of all cytokine concentrations (except TNF-α) and CBPI $_{pain}$, CBPI $_{interference}$ and CSOM, after $\log_{10}$ transformation of the data. Friedman test was used to analyze these associations for TNF-α which followed a non-normal distribution. When linear models were used, age, sex, and weight were considered as co-factors. For the associations with CBPI $_{pain}$, CBPI $_{interference}$ and CSOM, the control group was excluded because all data for CBPI were equal to zero and the CSOM was not part of the assessment of the control population.

## Results

### Animals

Three dogs were excluded for the following reasons: suspected immune-mediated disease of the central nervous system, mast cell tumor diagnosed on day 21 and significant serum levels of gabapentin measured during the placebo period (treatment error; Fig 2), respectively.

Twenty-nine dogs completed the study (mean age ± SD: 6.6 ± 3.0 years and mean body weight ± SD: 27.0 ± 18.5 kg; 21 males and 8 females) (Fig 2). Breeds included Bernese Mountain Dog (n = 6), Cavalier King Charles Spaniel (n = 5), Labrador Retriever (n = 4), Siberian Husky (n = 2), mixed-breed (n = 2), Poodle Toy (n = 1), Golden Retriever (n = 1), Polish Tatra Sheepdog (n = 1), Wire Fox Terrier (n = 1), Boxer (n = 1), Pug (n = 1), Longhaired Dachshund (n = 1), Basset Hound (n = 1), Beagle (n = 1) and Pomeranian (n = 1). Duration of pain prior to enrollment ranged from 1 to 60 months according to the owner's report with a median of 12 months. Pain-associated conditions diagnosed by MRI included spondylomyelopathies, lumbosacral syndromes, intervertebral disk disease with or without discospondylitis, Chiari malformations, congenital vertebral malformation, nerve sheath tumor and meningeal tumor. Dogs had at least one of the above lesions in the MRI. Dogs with NeuP were older than controls ($P$ = .021) but there was no difference for body weight ($P$ = .36). There were significantly more males in the NeuP group than in controls (72.4% versus 37.5%, $P$ = .030).

### Adverse reaction / analgesic failure

One dog developed erythema associated with pruritus shortly after the treatment with gabapentin-meloxicam. Clinical signs subsided after meloxicam was stopped. Owners reported a history of food allergy and it was believed that the erythema could be associated with the palatable agent contained in chewable tablets of meloxicam. Results for this treatment (gabapentin-meloxicam) were excluded from data analyses, and the dog started its placebo treatment (2nd block) immediately after. Treatment with gabapentin was administered later (3rd block) and data for placebo and gabapentin were included in the analyses. Other adverse effects were not recorded with the other treatment blocks and the dog completed the study. Analgesic failure was observed in one patient with nerve sheath tumor receiving gabapentin-meloxicam in the first block. This dog was excluded from the study but data from its initial presentation were included in statistical analysis. Finally, recurrence of severe signs of pain prompted a re-

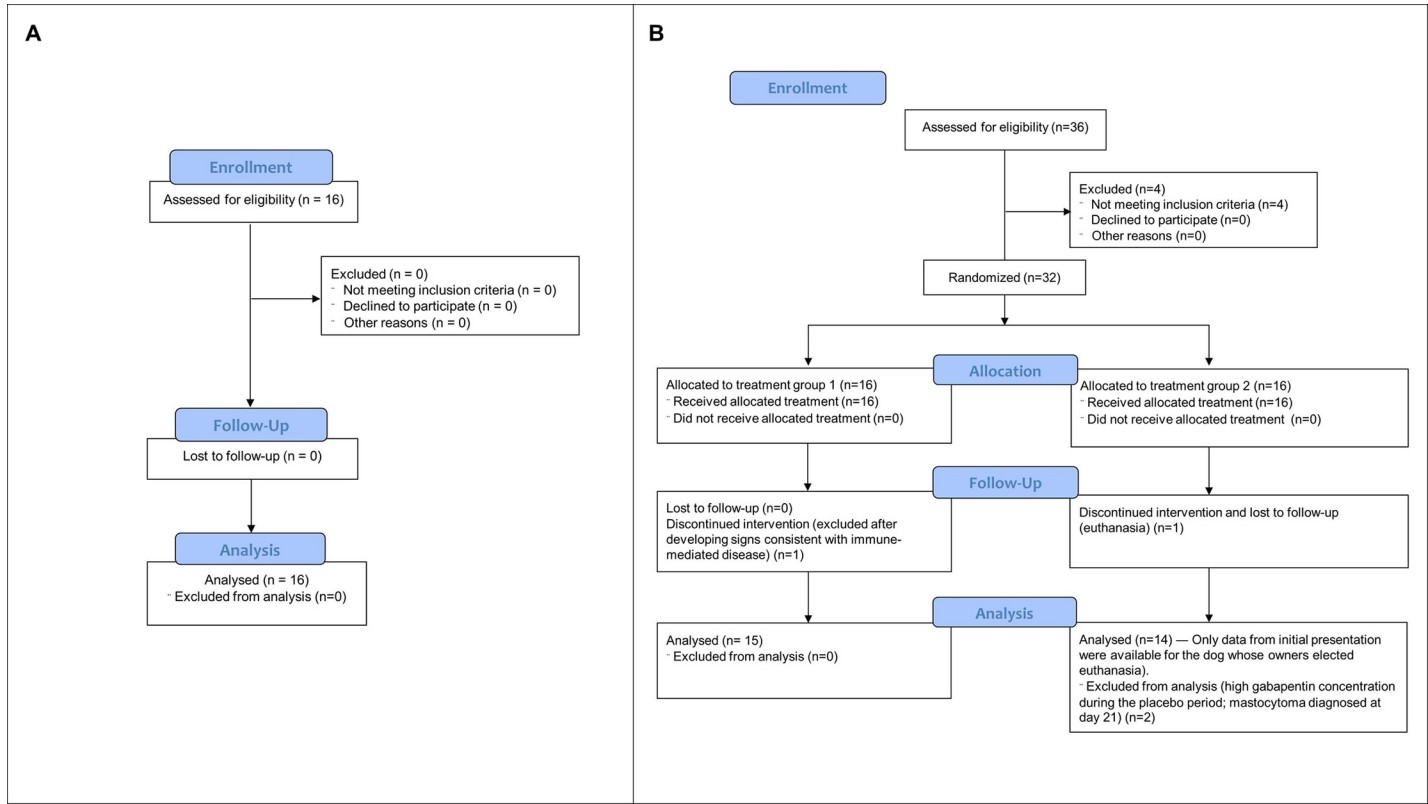

**Fig 2.** CONSORT Flow Diagram showing the flow of **a)** healthy dogs and **b)** dogs with neuropathic pain through the study.

evaluation in one individual with osseous-associated cervical spondylomyelopathy after 4 days into the placebo period.

## Quantitative sensory testing

Mean ± SEM MNT and ENT did not differ between healthy controls and NeuP at initial presentation (MNT: 10.4 ± 0.8 and 10.6 ± 0.6; $P$ = .86 and ENT: 49.5 ± 6.7 and 48.8 ± 5.2; $P$ = .94, respectively). There was an effect of body weight on both modalities (MNT: $P$ < .0001; ENT: $P$ = .0055) with higher thresholds observed in heavier dogs.

Mean ± SEM ΔMNT was significantly larger in healthy controls than in NeuP (2.3 ± 0.9 N and -0.2 ± 0.7 N, respectively; $P$ = .045). Body weight ($P$ = .47), sex ($P$ = .88) and age ($P$ = .076) were not associated with ΔMNT.

Treatment order did not influence ENT and MNT ($P$ = .20 and $P$ = .80, respectively). In NeuP, ENT, MNT or ΔMNT were not affected by treatment ($P$ = .06, $P$ = .94 and $P$ = .21, respectively), and there was no association between ENT, MNT, ΔMNT and sex ($P$ = .22, $P$ = .90 and $P$ = .99) or age ($P$ = .12, $P$ = .76 and $P$ = .25), respectively. Both ENT and MNT were positively associated with body weight (p < .0001) but not ΔMNT ($P$ = .50) (Table 2).

The percentage of positive and negative ΔMNT was calculated for each group (healthy controls and NeuP) and after each treatment block. In healthy controls, 33.3% of the dogs had a negative ΔMNT (i.e. facilitatory profile) whereas 66.7% showed a positive ΔMNT (i.e. inhibitory profile) (Fig 3). The percentage of negative ΔMNT were as follows in NeuP: 61.5% of dogs had a negative ΔMNT at initial presentation, 34.6% after gabapentin, 53.8% after gabapentin-meloxicam and 63.0% after placebo; positive ΔMNT was recorded in 38.5% of NeuP at initial

**Table 2. Electrical and mechanical nociceptive thresholds (ENT and MNT, respectively) and changes in mechanical nociceptive thresholds after application of a conditioning stimulus (ΔMNT) in dogs with naturally-occurring presumptive neuropathic pain before and after each treatment period.**

|  | ENT (mA) | MNT (N) | ΔMNT (N) |
|---|---|---|---|
| **Baseline** | 49.5 ± 3.4 (n = 29) | 10.2 ± 0.5 (n = 29) | - 0.1 ± 0.6 (n = 27) |
| **Placebo** | 42.3 ± 3.4 (n = 28) | 10.3 ± 0.5 (n = 28) | - 0.9 ± 0.6 (n = 26) |
| **Gabapentin** | 38.3 ± 3.4 (n = 28) | 10.1 ± 0.5 (n = 28) | 0.8 ± 0.6 (n = 26) |
| **Gabapentin-meloxicam** | 39.7 ± 3.4 (n = 28) | 10.3 ± 0.5 (n = 28) | 0.5 ± 0.6 (n = 26) |

Data shown as mean ± SEM after a mixed linear model to analyze ENT, MNT and ΔMNT with treatment as the main effect and sex, age and body weight as covariates.

presentation, 65.4% after gabapentin, 46.2% after gabapentin-meloxicam and 37.0% after placebo (Fig 3).

## Pain assessment tools

The cumulative score for the CBPI severity and interferences domains were 0 for all control dogs. The CBPI $_{overall\ impression}$ for these dogs ranged from very good (n = 2) to excellent (n = 14). The median (range) scores for CMPS-SF for control dogs were 0 (0–1) and were 5 (0–9) for NeuP.

The treatment order for NeuP did not significantly change the scores of CSOM ($P$ = .07), CBPI $_{pain}$ ($P$ = .064), CBPI $_{interference}$ ($P$ = .15) and CMPS-SF ($P$ = .58). There was no association between sex and age for CSOM ($P$ = .94 and $P$ = .42, respectively), CBPI $_{pain}$ ($P$ = .97 and $P$ = .80, respectively) and CBPI $_{interference}$ ($P$ = .81 and $P$ = .28, respectively).

*CSOM* − Treatment influenced CSOM scores ($P$ < .0001). Higher scores (more difficult to perform a given activity) were attributed by owners at presentation than after each treatment including placebo (Table 3).

*CBPI $_{pain}$* − Treatment influenced CBPI $_{pain}$ ($P$ = .002). These scores were higher (more painful) at presentation than after gabapentin or gabapentin-meloxicam (Table 3).

*CBPI $_{interference}$* − Treatment influenced CBPI $_{interference}$ ($P$ = .02). These scores were higher at presentation (locomotion more severely affected) than after gabapentin-meloxicam (Table 3).

*CBPI $_{overall\ impression}$* − Treatment influenced CBPI $_{overall\ impression}$ ($P$ = .0002). These scores were higher (improved overall impression) after gabapentin than at presentation (Table 3).

*CMPS-SF* − Treatment influenced CMPS-SF scores ($P$ = .002). These scores were higher at presentation than after gabapentin and gabapentin-meloxicam and were higher after placebo than gabapentin-meloxicam (Table 3). Pain scores were higher in male than female dogs ($P$ = .038).

## Serum concentrations of gabapentin and inflammatory cytokines

Mean ± SD dose of gabapentin was 11.05 ± 1.46 mg/kg (range: 8.62–14.49 mg/kg). Most of the dogs included in this study had undetectable concentrations of gabapentin at presentation and at day 14 (end of placebo period); minimal concentrations of gabapentin were found in the serum of 5 dogs at presentation (≤ 0.11 μg/mL; four had received a dose of gabapentin 24 to 48 hours before blood drawn) and 4 dogs at day 14 (< 0.26 μg/mL, except for one dog that had concentrations of approximately 9 μg/mL and was excluded from analysis). Concentrations of gabapentin in the first and third blocks ranged from 0.36–18.47 μg/mL. Mean concentrations of gabapentin ± SD were 8.53 ± 3.07 μg/mL and 7.13 ± 5.09 μg/mL after gabapentin alone or in combination with meloxicam, respectively.

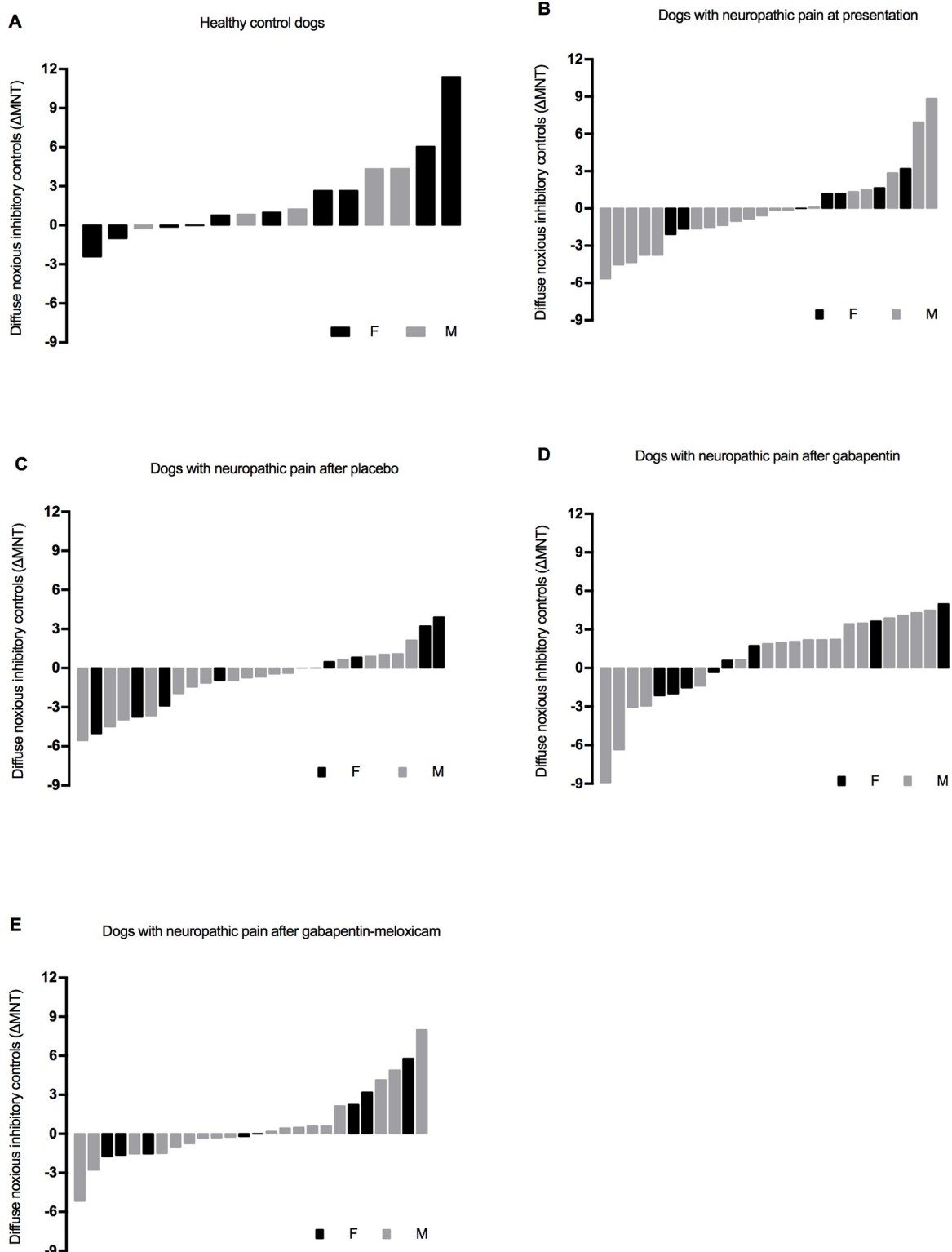

**Fig 3.** Diffuse Noxious Inhibitory Control (DNIC) in the population of a) healthy dogs, b) dogs with neuropathic pain at initial presentation, c) after placebo, d) after gabapentin-meloxicam and e) after gabapentin alone. Negative values represent facilitatory while positive values represent inhibitory conditioned pain modulation.

**Table 3. Pain scores obtained in dogs with naturally-occurring neuropathic pain before and after each treatment period.** Data are presented as mean ± SEM for scores from Client Specific Outcome Measures (CSOM), Canine Brief Pain Inventory (CBPI $_{pain}$ and CBPI $_{interference}$), and short-form Glasgow Composite Measure Pain Scale (CMPS-SF). Data are presented as median (range) for scores from CBPI $_{overall\ impression}$.

| | CSOM | CBPI<br>pain | CBPI $_{interference}$ | CBPI<br>overall impression | CMPS-SF |
|---|---|---|---|---|---|
| **Baseline** | 10.4 ± 0.7 (n = 25) | 20.2 ± 1.8 (n = 28) | 21.2 ± 1.8 (n = 28) | 2.0 (1.0–4.0) (n = 29) | 4.4 ± 0.5 (n = 24) |
| **Placebo** | **8.5 ± 0.7** (n = 24) | 17.9 ± 1.8 (n = 27) | 17.0 ± 1.8 (n = 27) | 2.8 (1.0–5.0) (n = 28) | 3.9 ± 0.5 (n = 19) |
| **Gabapentin** | **7.7 ± 0.7** (n = 20) | **15.7 ± 1.9** (n = 23) | 16.4 ± 1.9 (n = 22) | **3.0 (2.0–5.0)** (n = 24) | **2.9 ± 0.5** (n = 18) |
| **Gabapentin-meloxicam** | **7.5 ± 0.7** (n = 24) | **14.7 ± 1.9** (n = 24) | **16.6 ± 1.9** (n = 24) | 3.0 (1.0–5.0) (n = 24) | **2.5 ± 0.5**\* (n = 18) |

Data in bold are significantly different from results at initial presentation and the asterisk

(\*) marks significant difference compared with placebo.

Standard measure obtained for MCP-1 on one of the two plates used for the analysis was not included in the quality control range provided by the manufacturer therefore, corresponding data for MCP-1 were excluded. Two analytes (IFN-γ and IL-2) showed a proportion of results below detection level (out of range) superior to 50% and were therefore not analyzed. Among the population studied, 7 dogs were excluded from the cytokine analyses (chronic skin conditions: n = 4; oral inflammatory disease: n = 2; femoro-tibial effusion: n = 1). Concentrations of cytokines measured in controls and NeuP before treatment are summarized in Table 4. No differences were found between groups. Significant effects of sex and body weight were found for some analytes (Tables 4 and 5). A significant correlation was found between MCP-1 concentrations and the overall impression of the owners on their dogs' quality of life (Tables 6 and 7).

## Discussion

This study provides novel insights on the sensory profile and pain burden of dogs with naturally-occurring NeuP undergoing medical treatment. The functional assessment of DNIC in dogs with NeuP showed that ΔMNT remained mostly unchanged or even decreased (i.e. negative values, indicating a facilitatory profile) after the application of a conditioning stimulus.

**Table 4. Cytokine concentrations (median and range) in pg/mL measured in healthy control dogs and in dogs with presumptive neuropathic pain (NeuP) using the Milliplex Canine Cytokine Panel.**

| | Controls | NeuP | p | Covariates effect | | |
|---|---|---|---|---|---|---|
| | n = 13; MCP-1: n = 11 | n = 23; MCP-1: n = 11 | | P $_{sex}$ | P $_{age}$ | P $_{weight}$ |
| **GM-CSF** | 15.02 (0.56–219.95) | 30.12 (0.56–240.47) | 0.53 | 0.17 | 0.90 | 0.18 |
| **KC-like** | 417.23 (203.88–1,391.12) | 668.54 (67.69–1,381.57) | 0.54 | 0.56 | 0.34 | 0.56 |
| **IP-10** | 7.00 (1.42–34.35) | 7.79 (0.65–62.87) | 0.16 | 0.51 | 0.14 | 0.49 |
| **IL-6** | 6.16 (2.02–80.89) | 8.79 (1.89–78.55) | 0.15 | **0.015** | 0.67 | 0.06 |
| **IL-7** | 34.34 (3.36–187.41) | 21.50 (1.11–133.66) | 0.07 | 0.10 | 0.85 | 0.18 |
| **IL-8** | 2,504.34 (966.25–3,768.76) | 3,311.17 (690.87–13,131.05) | 0.35 | 0.75 | 0.11 | 0.47 |
| **IL-10** | 0.94 (0.33–162.04) | 1.53 (0.33–44.96) | 0.42 | 0.10 | 0.36 | 0.61 |
| **IL-15** | 47.85 (7.24–2,381.73) | 47.85 (4.98–1,251.31) | 0.33 | 0.59 | 0.62 | **0.013** |
| **IL-18** | 25.32 (10.71–178.37) | 24.15 (8.92–141.83) | 0.06 | 0.17 | 0.49 | 0.19 |
| **MCP-1** | 205.98 (154.27–410.62) | 259.17 (174.39–539.18) | 0.52 | 0.41 | 0.07 | 0.10 |
| **TNFα** | 1.25 (0.05–59.87) | 1.63 (0.05–43.02) | 0.74 | NA | NA | NA |

NA = Data non available (nonparametric test). Data in bold are significant.

**Table 5. Cytokine concentrations (median and range) in pg/mL measured in dogs with presumptive neuropathic pain (NeuP) before and after treatments of placebo, gabapentin, gabapentin-meloxicam using the Milliplex Canine Cytokine Panel.**

| | Baseline | Placebo | Gabapentin | Gabapentin-meloxicam | p | Covariates effect | | |
|---|---|---|---|---|---|---|---|---|
| | n = 23; MCP-1: n = 11 | n = 22; MCP-1: n = 11 | n = 22; MCP-1: n = 11 | n = 20; MCP-1: n = 11 | | p sex | p age | p weight |
| GM-CSF | 30.12 (0.56–240.47) | 20.74 (0.56–265.78) | 35.16 (0.56–336.65) | 16.81 (0.56–262.01) | 0.73 | 0.45 | 0.78 | 0.06 |
| KC-like | 668.54 (67.69–1,381.57) | 589.36 (80.65–1,596.31) | 492.40 (41.48–1,520.10) | 564.58 (46.36–1,570.21) | 0.38 | 0.96 | 0.31 | 0.29 |
| IP-10 | 7.79 (0.65–62.87) | 5.96 (0.65–34.60) | 6.27 (0.65–37.67) | 7.32 (0.65–43.69) | 0.73 | 0.96 | 0.17 | 0.25 |
| IL-6 | 8.79 (1.89–78.55) | 6.58 (2.02–86.76) | 12.16 (2.35–100.41) | 7.39 (2.35–79.68) | 0.57 | 0.06 | 0.56 | **0.035** |
| IL-7 | 21.50 (1.11–133.66) | 16.03 (1.11–149.90) | 18.66 (1.98–172.46) | 13.98 (1.11–141.01) | 0.25 | **0.048** | 0.99 | 0.10 |
| IL-8 | 3311.17 (690.87–13,131.05) | 3,462.72 (450.80–9,539.46) | 3,335.68 (1,080.60–19,188.58) | 3,276.29 (889.47–10,406.34) | 0.99 | 0.78 | 0.06 | 0.99 |
| IL-10 | 1.53 (0.33–44.96) | 2.53 (0.33–44.96) | 2.09 (0.33–75.93) | 0.95 (0.33–51.84) | 0.49 | 0.08 | 0.15 | **0.015** |
| IL-15 | 47.85 (4.98–1,251.31) | 21.05 (4.98–1,255.93) | 48.06 (4.98–1,431.35) | 32.11 (4.98–1,302.06) | 0.52 | 0.72 | 0.64 | **0.001** |
| IL-18 | 21.15 (8.92–141.83) | 20.96 (9.49–158.79) | 22.23 (8.92–186.90) | 20.69 (7.71–149.13) | 0.17 | **0.032** | 0.33 | **0.006** |
| MCP-1 | 259.17 (174.39–539.18) | 261.84 (176.16–409.52) | 253.67 (159.41–401.28) | 250.84 (163.47–492.68) | 0.91 | 0.08 | **0.048** | 0.40 |
| TNF α | 1.63 (0.05–43.02) | 0.92 (0.05–48.27) | 2.30 (0.05–57.41) | 0.29 (0.05–44.18) | 0.23 | NA | NA | NA |

A nonparametric test was used to analyze TNF α, therefore it was not possible to test for the effect of sex, age and weight on the concentration of this analyte (NA = non applicable). Data in bold are significant.

These values were significantly different than healthy controls that presented mean positive values for ΔMNT (i.e. inhibitory profile) [10]. This result suggests a dysfunctional DNIC in dogs with NeuP, which is consistent with previous results obtained by different methods of DNIC assessment in dogs suffering from osteoarthritis [17], osteosarcoma [18] and in rodent models of NeuP [19]. Therefore, NeuP may present changes in the descending modulatory mechanisms of pain (facilitatory over inhibitory input) reinforcing the need for disease-modifying therapies that produce changes in central pain modulation (e.g. gabapentinoids). The assessment of DNIC using the percentage of positive and negative ΔMNT has been described in humans with fibromyalgia [11]. Following the activation of spinal cord neurons conveying nociceptive input, supraspinal descending controls are normally activated to produce an inhibitory effect at the level of the dorsal horn of the spinal cord. In healthy conditions, the expected outcome would be the attenuation of subsequent painful input [20]. Therefore, animals with a

**Table 6. Results of the statistical analysis evaluating the association between cytokines concentrations and a) owners' perception of their dog's quality of life b) CMPS-SF.**

| | CBPI overall impression (n = 36) | | CMPS-SF (n = 32) | |
|---|---|---|---|---|
| | Spearman's rho correlation coefficient | Significance (P value) | Spearman's rho correlation coefficient | Significance (P value) |
| GM-CSF | 0.056 | 0.74 | -0.027 | 0.87 |
| KC-like | -0.092 | 0.59 | -0.015 | 0.94 |
| IP-10 | 0.091 | 0.59 | -0.21 | 0.24 |
| IL-6 | -0.037 | 0.83 | 0.047 | 0.79 |
| IL-7 | 0.15 | 0.37 | -0.22 | 0.22 |
| IL-8 | -0.21 | 0.22 | 0.13 | 0.47 |
| IL-10 | -0.175 | 0.30 | 0.086 | 0.63 |
| IL-15 | 0.27 | 0.11 | -0.19 | 0.29 |
| IL-18 | 0.18 | 0.29 | -0.12 | 0.50 |
| MCP-1 | **-0.38** | **0.024** | 0.31 | 0.08 |
| TNF- α | 0.118 | 0.48 | -0.125 | 0.49 |

Data in bold are significant.

**Table 7. Results of the statistical analysis evaluating the association between cytokines concentrations and a) Client Specific Outcome Measures scores b) Canine Brief Pain Inventory (section pain) scores c) Canine Brief Pain Inventory (section interference, locomotion) scores.**

|  | CSOM |  | CBPI $_{pain}$ |  | CBPI $_{interference}$ |  |
|---|---|---|---|---|---|---|
|  | Slope (SEM) | p value | Slope (SEM) | p value | Slope (SEM) | p value |
| GM-CSF | 0.000489 (0.00915) | 0.96 | -0.00143 (0.00389) | 0.71 | 0.00155 (0.00322) | 0.63 |
| KC-like | -0.00072 (0.00644) | 0.91 | 0.00232 (0.00297) | 0.44 | -0.00069 (0.00242) | 0.78 |
| IP-10 | -0.00248 (0.00546) | 0.65 | 0.00087 (0.00281) | 0.76 | 0.000294 (0.0023) | 0.90 |
| IL-6 | -0.0102 (0.00973) | 0.30 | -0.00125 (0.00417) | 0.76 | 0.000642 (0.00353) | 0.86 |
| IL-7 | 0.00218 (0.00649) | 0.74 | -0.00116 (0.00321) | 0.72 | 0.000762 (0.00267) | 0.78 |
| IL-8 | 0.000556 (0.00998) | 0.96 | -0.00207 (0.00416) | 0.62 | 0.000965 (0.00356) | 0.79 |
| IL-10 | 0.000524 (0.0133) | 0.97 | 0.0028 (0.00585) | 0.63 | -0.00248 (0.00488) | 0.61 |
| IL-15 | -0.0151 (0.0131) | 0.25 | -0.00844 (0.00525) | 0.11 | -0.00454 (0.00445) | 0.31 |
| IL-18 | -0.00225 (0.00499) | 0.65 | -0.00159 (0.00221) | 0.47 | 0.00082 (0.00186) | 0.66 |
| MCP-1 | -0.00367 (0.00418) | 0.39 | 0.000263 (0.00203) | 0.90 | -0.00082 (0.00171) | 0.63 |
| TNF-α | 0.0202 (0.03) | 0.50 | 0.00252 (0.0107) | 0.82 | 0.00385 (0.01) | 0.70 |

functional DNIC should show positive values of ΔMNT (i.e. inhibitory profile) after the application of a conditioning stimulus. Indeed, most of the healthy individuals showed an inhibitory profile. However, approximately a third of this population had ΔMNT negative values (i.e. facilitatory profile). Similar findings have been reported in healthy dogs and humans [11,18]. In this study, approximately 60% of dogs with NeuP had a facilitatory profile at presentation and after the administration of placebo, with an approximate 2-fold increase when compared with the percentage of healthy dogs with the same sensory profile. On the other hand, the percentage of dogs with facilitatory profile after gabapentin was comparable with healthy controls. A similar effect has been found with pregabalin in human patients with fibromyalgia [21]. This finding is consistent with recent research showing an activation of the inhibitory system by increased activity of noradrenergic neurons located in the locus coeruleus after the administration of gabapentin [22]. In our study, the DNIC function of NeuP was regained after gabapentin. It is not clear why the same effect was not observed after the administration of gabapentin-meloxicam where approximately 50% of NeuP continued to show a facilitatory profile. In humans, neuropathic pain is known to show a low response to conventional therapies, including non-steroidal anti-inflammatory drugs [23]. However, despite being not statistically significant, there was a trend for ΔMNT values to be negative at presentation and after placebo, and positive after gabapentin and gabapentin-meloxicam. While DNIC and stress-induced analgesia are two endogenous analgesic mechanisms that can be triggered by a noxious stimulus [24], the authors used a fear-free approach to minimize stress-induced analgesia and we believe the results are indeed a reflection of DNIC profile of these patients.

Central sensitization has been observed in patients with NeuP [25]. In animal models of NeuP based on peripheral nerve injury, this phenomenon is commonly studied by measuring nociceptive thresholds in a remote area from the injury [26]. For this reason, it was deemed that using the 'less affected limb' for the assessment of the DNIC would provide a more accurate value than using the 'most affected limb'. Also, ENT and MNT measured at the affected, but also other limbs were averaged for each individual. Thresholds were expected to be overall lower in NeuP than in controls due to potential for central sensitization. However, MNT and ENT were not significantly different between the two populations and did not change after treatments in NeuP. This could be explained by the great individual variability of both QST modalities in dogs from different breeds, ages and body weight [10]. On the other hand, a recent study investigating NeuP in Cavalier King Charles Spaniels dogs reported higher MNT

after the administration of pregabalin when compared with baseline or placebo treatment [27]. The different findings could rely on the homogeneity of the population studied (same breed and same underlying disease), different testing sites, technique or nociceptive threshold device. Finally, both ENT and MNT were influenced by body weight. A positive correlation between body weight and MNT has been described in healthy dogs [28]. Since our two populations (controls and NeuP) had similar body weight, this was not considered as a confounding factor in the present study.

The pain burden caused by NeuP in dogs was evaluated at presentation and after therapy using different pain scoring systems. The CBPI allowed the evaluation of NeuP in terms of comfort (CBPI $_{pain}$), function (CBPI $_{interference}$) and quality of life (CBPI $_{overall\ impression}$). The function was further assessed using the CSOM. These two methods of pain assessment (CBPI and CSOM) were used to investigate the pain burden in a familiar environment as perceived by owners who were masked to the treatment. A method of acute pain assessment (CMPS-SF) was used for the veterinarian's evaluation due to the possibility of an acute episode of pain related to the chronic underlying condition and the lack of valid pain assessment instruments to evaluate NeuP in dogs. Gabapentin alone or in combination with meloxicam reduced pain scores when compared with presentation, but not placebo, using the CSOM, CBPI $_{pain}$ and CMPS-SF. Gabapentin exerts its analgesic effect through its action on supraspinal region to promote descending inhibition of nociceptive stimuli [22], and it binds to the $\alpha_2$-$\delta$ subunit of the voltage-gated calcium channels involved in the maintenance of mechanical hypersensitivity in rodent models of NeuP [29]. The CBPI $_{overall\ impression}$ showed an improved quality of life after the administration of gabapentin when compared with presentation. The same results were not observed for gabapentin-meloxicam. However, less than one third of dogs were classified with a "poor" or "fair" quality of life after gabapentin or gabapentin-meloxicam, whereas at least 50% of dogs were classified within these categories after placebo and at presentation. The combination of gabapentin and meloxicam was associated with improved activity using CBPI $_{interference}$ when compared with presentation, and when using CMPS-SF compared with placebo. However, mean values for CBPI $_{interference}$ between gabapentin and gabapentin-meloxicam groups were similar, and it is difficult to know the clinical relevance of these findings. This study could not determine in which patients the administration of meloxicam would be beneficial in combination with gabapentin. Meloxicam is a non-steroidal anti-inflammatory drug, a preferential cyclooxygenase 2 (COX-2) inhibitor, used for the treatment of osteoarthritis in dogs [30]. An overexpression of COX-2 has been observed with peripheral NeuP and one of the reasons meloxicam was used in this study [31].

Resting was recommended as part of treatment and could have contributed to pain relief in this study. Additionally, a carry-over effect after the first week of treatments (gabapentin or gabapentin-meloxicam) cannot be ruled out especially considering the low concentrations of gabapentin detected on day 14 at the end of placebo administration. However, a significant effect was not observed for treatment order and it is unlikely that these small serum concentrations of gabapentin would produce an analgesic effect in dogs with NeuP. A significant improvement was found after placebo treatment using the CSOM. Additionally, CBPI and CMPS-SF pain scores after treatment with gabapentin or gabapentin-meloxicam were significantly improved when compared with baseline, but not after placebo. Therefore, the analgesic effects of gabapentin or gabapentin-meloxicam could be debatable, if one considers only a positive outcome when treatments are superior than placebo. However, CBPI scores (where owners were masked to treatment) were not significantly different between initial presentation and placebo; these findings could suggest a beneficial effect of the active treatments. Gabapentin or gabapentin-meloxicam may have an effect in the clinical setting, even if it may have been biased by a placebo effect in this study. The latter has been reported in veterinary clinical trials

including chronic painful conditions. Indeed, the CSOM and the CBPI could have been biased by the so-called "caregiver placebo effect", since these instruments involve proxy measures of pain [32]. It is believed that time spent with the patient, better care (compliance with treatment administration), empathy, optimism and desire for the treatment to work could improve caregiver evaluations after placebo and make it difficult to show superiority of an efficacious medication to placebo [33]. The caregiver placebo effect is related to improved ratings of subjective outcomes (pain scores) in the absence of improvement in objective measures [33]. On the other hand, when using DNIC facilitatory profiles as means of objective assessment, placebo and at initial presentation had similar ΔMNT profiles which is approximately twice more than dogs with NeuP treated with gabapentin. These results highlight how difficult chronic pain assessment in companion animals can be especially when validated tools specific for the assessment of NeuP are not available. Finally, the veterinarian performing evaluations was masked to the first and third blocks (gabapentin or gabapentin-meloxicam), but not the second (placebo) block of treatments. Thus, the evaluation of the dogs after placebo treatment relied mostly on the unbiased owners' evaluation.

In the present study, serum concentrations of gabapentin were evaluated as an indirect assessment of owners' compliance to treatment administration and to report these concentrations for *posteriori* studies potentially correlating therapeutic levels with dosage regimens, sex, breed, age and the analgesic efficacy of gabapentin. The concentrations of gabapentin required to alleviate NeuP remain unknown. Based on pharmacologic modelling, the potency of gabapentin (EC 50) in rats for its anti-allodynic effect was reported between 1.4 to 16.4 μg/mL [34,35] and 5.35 μg/mL for the treatment of neuropathic pain in man [36]. In our study, dogs had concentrations ranging between 0.36 and 18.5 μg/mL but timing of blood collection could not be standardized due to owners' constraints for scheduling re-evaluations and time of drug administration. Given both veterinarian's and owners' positive outcomes, the dosage regimens for gabapentin were considered effective in the treatment of NeuP in dogs. However, there was a large range of concentrations showing significant individual variability that could impact the pharmacokinetics and potentially the pharmacodynamics of the drug in the clinical setting.

The concentrations of inflammatory cytokines measured in this study are consistent with previously published data in healthy dogs [37], with large individual concentration variability, especially considering individuals of different breeds and suffering from different neurological pathologies. Therefore, the lack of significant differences between control and NeuP groups, or between treatments in this study may reflect a type 2 error, more than an actual homogeneity of these populations. A higher concentration of MCP-1 was associated with a worse appreciation of the quality of life of their dog by the owner. These results corroborate previous findings in humans where MCP-1 concentrations were positively associated with more severe fibromyalgia-related pain when evaluated with the brief pain inventory [38]. Our results also suggest that future investigations on inflammatory cytokines in canine NeuP should divide the population into subgroups based on sex and body weight to better understand the disease.

The limitations of our study design including a partially masked evaluator and a bias towards a caregiver placebo effect have been discussed. Due to ethical considerations in clinical pain research, dogs experiencing pain were immediately treated either before (administration of remifentanil) or during the study (rescue analgesia), therefore introducing a potential bias in the results. However, in the present study, these interventions were minimal (exclusion during the first block with gabapentin-meloxicam, n = 1; four days of placebo period instead of 7, n = 1) but it may have contributed to a mild overall improvement observed after placebo or gabapentin. The initial assessment may also have been altered by the administration of remifentanil in two dogs before the withdrawal period of 60 minutes. The drug may have provided sustained analgesia reducing clinical signs of central sensitization in dogs with NeuP before

QST at initial presentation. Also, there is no definitive test to diagnose NeuP. Therefore, inclusion criteria were determined to meet the most recent definition of NeuP by the International Association for the Study of Pain: "pain arising as a direct consequence of a lesion or disease affecting the somatosensory system". All dogs included had a long-term history of pain and a confirmed neurological lesion found at MRI. Additionally, most dogs had delayed paw placements or ataxia which indicated an involvement of the somatosensory system. Recognition of NeuP remains a challenge in veterinary medicine and in non-verbal human patients since it is characterized by the combination of sensory qualities that can only be self-reported [39].

In conclusion, dogs with NeuP have changes in sensory profile characterized by a dysfunctional (deficient) DNIC compared with healthy controls. These results could be the expression of maladaptive changes in favor of pain facilitation over inhibition in the central pain processing. The percentage of dogs with facilitatory sensory profile was similar at baseline and after placebo, and between controls and after gabapentin, but not gabapentin-meloxicam, suggesting that gabapentin alone may have improved DNIC. According to CBPI (masked owners' assessment) and CMPS-SF (non-masked veterinarian's assessment), pain burden was reduced after gabapentin and/or gabapentin-meloxicam when compared with initial presentation. However, these scores were not significantly different than placebo, nor placebo was superior to baseline evaluations, with the exception of CSOM scores. Resting as part of treatment may have helped with decreased pain scores during the study. Results may have been biased by a caregiver placebo effect and other study limitations including the low number of animals enrolled, population heterogeneity including a variety of diseases, and the lack of a validated pain assessment tool for canine NeuP. Inflammatory cytokines were not different between groups or treatments. More studies on canine NeuP are warranted to determine best therapeutic regimens for the disease.

## Supporting information

**S1 File. Serum concentrations of gabapentin in dogs.**
(DOCX)

**S1 Database.**
(XLSX)

## Acknowledgments

The authors would like to thank Fleur Gaudette from the Pharmacokinetics core facility of the Centre de Recherche, Centre hospitalier de l'Université de Montréal (CRCHUM) for carrying out LC-MS/MS method development, validation, and sample analysis and the dedicated pet owners who participated to this study.

## Author Contributions

**Conceptualization:** Hélène L. M. Ruel, Paulo V. Steagall.

**Data curation:** Hélène L. M. Ruel, Ryota Watanabe, Paulo V. Steagall.

**Formal analysis:** Hélène L. M. Ruel, Guy Beauchamp, Jean-Philippe Auger.

**Funding acquisition:** Hélène L. M. Ruel, Paulo V. Steagall.

**Investigation:** Hélène L. M. Ruel, Jean-Philippe Auger, Mariela Segura, Paulo V. Steagall.

**Methodology:** Hélène L. M. Ruel, Ryota Watanabe, Marina C. Evangelista, Guy Beauchamp, Jean-Philippe Auger, Paulo V. Steagall.

**Project administration:** Paulo V. Steagall.

**Resources:** Ryota Watanabe, Marina C. Evangelista, Jean-Philippe Auger, Mariela Segura.

**Software:** Guy Beauchamp.

**Supervision:** Mariela Segura, Paulo V. Steagall.

**Validation:** Hélène L. M. Ruel, Ryota Watanabe, Marina C. Evangelista, Jean-Philippe Auger, Mariela Segura.

**Writing – original draft:** Hélène L. M. Ruel.

**Writing – review & editing:** Hélène L. M. Ruel, Ryota Watanabe, Marina C. Evangelista, Guy Beauchamp, Jean-Philippe Auger, Mariela Segura, Paulo V. Steagall.

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
