## [Decision Letter · Decision Letter 0]

18 Sep 2020

PONE-D-20-22054

Pain burden, sensory profile and inflammatory cytokines of dogs with naturally-occurring neuropathic pain treated with gabapentin alone or with meloxicam

PLOS ONE

Dear Dr. Steagall,

Thank you for submitting your manuscript to PLOS ONE. After careful consideration, we feel that it has merit but does not fully meet PLOS ONE’s publication criteria as it currently stands. Therefore, we invite you to submit a revised version of the manuscript that addresses the points raised during the review process.

We look forward to receiving your revised manuscript.

Kind regards,

Tamil Selvan Anthonymuthu, Ph. D

Academic Editor

PLOS ONE

Journal Requirements:

2.Thank you for stating the following in the Competing Interests section:

[This study was supported by Boehringer Ingelheim (Canada) Ltd and MITACS through the Mitacs Accelerate Program, and the American Kennel Club Canine Health Foundation.].

Reviewers' comments:

Reviewer's Responses to Questions

**Comments to the Author**

1. Is the manuscript technically sound, and do the data support the conclusions?

Reviewer #1: Partly

2. Has the statistical analysis been performed appropriately and rigorously? 

Reviewer #1: No

3. Have the authors made all data underlying the findings in their manuscript fully available?

Reviewer #1: Yes

4. Is the manuscript presented in an intelligible fashion and written in standard English?

Reviewer #1: Yes

5. Review Comments to the Author

Reviewer #1: PONE-D-20-22054

This randomized, cross-over, placebo-controlled study attempts to characterize the sensory profile and pain burden, as well as response to gabapentin and meloxicam therapy, in client-owned dogs diagnosed with presumptive neuropathic pain (NeuP) associated with a lesion of the nervous system. Sensory profile and pain burden were determined at presentation and after a 7-day therapy with placebo, gabapentin alone, or gabapentin+meloxicam. Gabapentin treatments were administered in random order but placebo was always the second treatment. Sensory profiles were determined by measuring the response thresholds to electrical and mechanical stimuli in all four limbs. Descending noxious inhibitor control (DNIC) was assessed as the difference between mechanical withdrawal thresholds before and after a conditioning stimulus consisting of applying pressure using an inflatable cuff (200 mmHg) for 60 seconds. The authors have previously validated the methodology. Pain burden was assessed with the use of 3 distinct clinical metrology instruments, none of which have been validated for assessing NeuP in dogs. Serum concentrations of a panel of cytokines and of gabapentin were determined. Sensory profiles and pain burden outcomes were compared to results obtained from a separate group of healthy dogs. Results showed that sensory thresholds in NeuP dogs were similar to those of heathy dogs, and they were not affected by any of the treatments. However, DNIC was significantly impaired in NeuP dogs compared to healthy controls, and this was normalized by gabapentin, but not gabapentin+meloxicam or placebo, therapy. Results of pain burden varied depending on the clinical instrument used. Plasma cytokine levels were not significantly different between NeuP and healthy controls and there were no overall effects of treatment. The authors concluded that “Dogs with NeuP have deficient inhibitory pain mechanisms. Pain burden was reduced after gabapentin and gabapentin-meloxicam depending on the pain scoring instrument used”.

This is an overall well-designed study with the use of placebo and blinding. It is an extremely important research topic with relevant findings. My concerns with the study are listed below.

- Major concerns:

The main concern with the study is the assessment of pain burden using pain scoring instruments that were not validated for assessing NeuP in dogs. The results are thus inconsistent across instruments, which preclude assertive conclusions to be made. Too much time is spent discussing the results obtained with these instruments and the resulting conclusion is highly questionable. However, the authors used validated methodology to assess DNIC, which is highly relevant in this study, and should be the guiding metric for discussing the results and final conclusions.

- Minor concerns:

A minor concern is the many variables/outcomes compared, which could lead to statistical inflation, and no mention of correction for multiple comparisons. Other/specific concerns are listed below:

Line 134: Were the capsules/tablets rounded to the nearest whole capsule or ½ tablet? You state they were “available” but were they actually administered as such? Please clarify.

Line 260: Why did you log-transformed the data prior to comparison?

Line 261: Which software was used for statistical analyzes?

Line 262: How did you determine symmetry?

Line 269: The explanation for pooling the data is not clear. If you pooled NeuP and control data, then you have a single data set and nothing to compare against… please clarify.

Line 273: So, the log transformation was carried out to normalize the non-normally distributed data, but TNF-a concentrations continued to be non-normally distributed after log transformation? Please clarify.

Figure 2: Poor figure quality. Impossible to read.

Line 290: You may wish to present the dog breeds in descending order according to n. Were the control dogs matched for breed? What were their breed?

Line 304: Was this dog with history of food allergies excluded?

Line 338-342: Were these differences in delta-MNT statistically significant?

Line 437-438: Unfortunately pain burden results are not convincing due to the dependency on instruments that have not been validated for NeuP.

Line 457: It is very interesting that meloxicam appeared to have negatively affected the improvement in delta-MNT obtained with gabapentin alone. You should try to discuss this further as there is anecdotal (at least) evidence that COX inhibitors may worsen neuropathic pain and certainly there is scientific evidence of their inefficacy against neuropathic pain.

Line 492-493: Some critical discussion is warranted on the significant decrease in CSOM scores during placebo treatment compared to baseline (Table 3), and regarding the lack of significance between placebo and treatments with all instruments (except the CMPS-SF for gabapentin/meloxicam).

Line 504-513: Although there are statistically significant differences, the data presented in Table 3 are not really convincing that meloxicam/gabapentin was superior to gabapentin alone. Their averages (and SEM and ranges) are almost identical, and the differences were primarily related to presentation values, not to placebo. Even for CMPS-SF, where gabapentin/meloxicam was significantly different than placebo (but the veterinarian was aware of placebo therapy) but gabapentin alone was not, the means (2.9 and 2.5) are essentially the same. Is a difference of 0.4 clinically relevant? Is this small difference worthwhile, especially considering the adverse effect profile of meloxicam? The argument that meloxicam could potentially have relieved pain from other conditions such as osteoarthritis is not entirely convincing either, since differences were not detected with CBPI, which was validated for dogs with osteoarthritis.

Line 514-515: In addition to the placebo effect, this is such an important aspect of this study that needs to be strongly considered in the discussion. Considering that there were no significant differences between placebo and both treatments, what your data really suggests is that resting+placebo would be an adequate therapy for NeuP in dogs. The significant difference between CMPS-SF and placebo can’t be entirely trusted as the veterinarian evaluating pain was aware of placebo treatment.

Line 539-540: The gabapentin treatments were no better than resting+placebo. Unfortunately your study was not able to show that gabapentin with our without meloxicam is better than rest/placebo in treating NeuP in dogs.

Line 577-578: The results of sensory profile (DNIC) are convincing but not the results obtained with the subjective instruments (CSOM, CBPI, CMPS-SF). The fact that results were dependent on the instrument used shows that they are not reliable for the purpose of this study and cannot be used to draw conclusions.

6. PLOS authors have the option to publish the peer review history of their article (what does this mean?). If published, this will include your full peer review and any attached files.

Reviewer #1: No

---

## [Author Response · Author response to Decision Letter 0]

16 Oct 2020

Response to Reviewers

Reviewer 1

- Major concerns:

The main concern with the study is the assessment of pain burden using pain scoring instruments that were not validated for assessing NeuP in dogs. The results are thus inconsistent across instruments, which preclude assertive conclusions to be made. Too much time is spent discussing the results obtained with these instruments and the resulting conclusion is highly questionable. However, the authors used validated methodology to assess DNIC, which is highly relevant in this study, and should be the guiding metric for discussing the results and final conclusions.

Answer: Thank you very much for these comments and suggestions. Unfortunately, there aren’t any clinical metrology instruments specifically for pain assessment in dogs with neuropathic pain. Hence, the study had to use other validated instruments for chronic pain assessment which often includes cases with neuropathic pain. 

The authors also feel important to highlight the discussion on DNIC. Indeed, the first two paragraphs of the discussion are dedicated to DNIC and facilitatory and inhibitory profiles in NeuP vs control dogs. One of these paragraphs explains the mechanisms behind DNIC and its evaluation with substantial information. For example, the authors compared their findings with the human literature. We tried to achieve a balanced discussion going over all the different aspects of the study (pain burden, sensory profile, inflammatory cytokines, placebo effect, concentrations of gabapentin and study limitations). We deleted some parts of the text while trying to have a more concise text. 

- Minor concerns:

A minor concern is the many variables/outcomes compared, which could lead to statistical inflation, and no mention of correction for multiple comparisons. Other/specific concerns are listed below: 

Answer: Unfortunately, this is one of the first studies involving a heterogenous population of dogs with neuropathic pain. There was a need to use different variables/outcomes to understand the effects of treatment, which as the study showed, was difficult to find. The original submission had mention of correction for multiple corrections (Tukey and Benjamini-Hochberg).

Line 134: Were the capsules/tablets rounded to the nearest whole capsule or ½ tablet? You state they were “available” but were they actually administered as such? Please clarify.

Answer: It has been now clarified.

Line 260: Why did you log-transformed the data prior to comparison?

Answer: Log-transformation was used to normalize data.

Line 261: Which software was used for statistical analyzes?

Answer: Data were analyzed using SAS (version 9.3; SAS Institute, Cary, NC, USA).

Line 262: How did you determine symmetry?

Answer: Our epidemiologist/statistician verified symmetry by visual inspection of data. 

Line 269: The explanation for pooling the data is not clear. If you pooled NeuP and control data, then you have a single data set and nothing to compare against… please clarify.

Answer: In this case, we were not making group comparisons but only looking at the Spearman correlation for CBPI overall impression and CMPS-SF scores and cytokine concentrations for the entire population.

Line 273: So, the log transformation was carried out to normalize the non-normally distributed data, but TNF-a concentrations continued to be non-normally distributed after log transformation? Please clarify.

Answer: For this variable, several values were at the detection limit and the log transformation was not helpful to achieve a more symmetrical distribution. This was the reason why we opted for the non-parametric test. 

Figure 2: Poor figure quality. Impossible to read.

Answer: The figure is slightly blurred in the final submission pdf provided by PlosOne, but the authors can read all the information. The figure is in accordance to PlosOne guidelines at 300 dpi. Unfortunately, the authors don’t know what the problem is.

Line 290: You may wish to present the dog breeds in descending order according to n. Were the control dogs matched for breed? What were their breed?

Answer: The order of dog breeds was changed and presented in descending order. Dog breeds, body weight and age for the control subjects have been added to the supplemental material. The control dogs could not be matched for breed; however, body weight was similar between groups.

Line 304: Was this dog with history of food allergies excluded?

Answer: Results for this treatment (gabapentin-meloxicam) were excluded from data analyses, and the dog started its placebo treatment (2nd block) immediately after. Treatment with gabapentin was administered later (3rd block) and data for placebo and gabapentin were included in the analyses.

Line 338-342: Were these differences in delta-MNT statistically significant?

Answer: Data for the % of positive and negative deltaMNT were not reported for significance, according to the original methodology (Potvin et al. 2016). However, a mixed linear model was used to analyze changes in deltaMNT as reported in the statistical analysis.

Line 437-438: Unfortunately pain burden results are not convincing due to the dependency on instruments that have not been validated for NeuP.

Answer: We appreciate the reviewer’s frustration; however, this shows how difficult pain assessment can be in chronic conditions such as neuropathic pain in the clinical setting, where pain scoring systems or tools have not been validated. Somehow, our knowledge on the subject has to move forward and we tried to come up with our best possible study design.

Line 457: It is very interesting that meloxicam appeared to have negatively affected the improvement in delta-MNT obtained with gabapentin alone. You should try to discuss this further as there is anecdotal (at least) evidence that COX inhibitors may worsen neuropathic pain and certainly there is scientific evidence of their inefficacy against neuropathic pain.

Answer: We have added a sentence on the inefficacy of NSAIDs in the treatment of neuropathic pain. However, we did not add any discussion on anecdotal evidences that may have little scientific value. 

Line 492-493: Some critical discussion is warranted on the significant decrease in CSOM scores during placebo treatment compared to baseline (Table 3), and regarding the lack of significance between placebo and treatments with all instruments (except the CMPS-SF for gabapentin/meloxicam).

Answer: The authors have expanded the discussion on the caregiver placebo effect with CSOM scores and the lack of significance between placebo and treatments.

Line 504-513: Although there are statistically significant differences, the data presented in Table 3 are not really convincing that meloxicam/gabapentin was superior to gabapentin alone. Their averages (and SEM and ranges) are almost identical, and the differences were primarily related to presentation values, not to placebo. Even for CMPS-SF, where gabapentin/meloxicam was significantly different than placebo (but the veterinarian was aware of placebo therapy) but gabapentin alone was not, the means (2.9 and 2.5) are essentially the same. Is a difference of 0.4 clinically relevant? Is this small difference worthwhile, especially considering the adverse effect profile of meloxicam? 

Answer: Agreed, and the discussion now reflects these findings. We no longer have any statement saying that gabapentin-meloxicam is superior to gabapentin. However, it is impossible to determine what clinical relevance is in the case of neuropathic pain in veterinary medicine which little is known, especially when similar improvements in %, for example, after treatment can be sufficient for pain relief in one patient, but not the other in a clinical trial.

The argument that meloxicam could potentially have relieved pain from other conditions such as osteoarthritis is not entirely convincing either, since differences were not detected with CBPI, which was validated for dogs with osteoarthritis.

Answer: The discussion has been rewritten providing the rationale of using an NSAID in this study.

Line 514-515: In addition to the placebo effect, this is such an important aspect of this study that needs to be strongly considered in the discussion. Considering that there were no significant differences between placebo and both treatments, what your data really suggests is that resting+placebo would be an adequate therapy for NeuP in dogs. The significant difference between CMPS-SF and placebo can’t be entirely trusted as the veterinarian evaluating pain was aware of placebo treatment.

Answer: The authors agreed that resting in the study provided means of pain relief as described in the discussion. However, the authors disagree that our results have shown that resting + placebo would be an adequate therapy for NeuP in dogs. Resting + placebo would have to be better than presentation, which is not the case.

Considering the limitations discussed in the manuscript such as challenges in pain assessment, the heterogenous population, number of patients with a variety of neuropathic painful conditions, among others, and mostly important a possible caregiver placebo effect, treatment effect cannot be only compared with placebo for means of analgesic efficacy. Scores at presentation should also be taken in consideration especially considering the caregiver placebo effect, which is now discussed in detail. 

Finally, the percentage of dogs with inhibitory and facilitatory profiles, according to ΔMNT (DNIC), were similar between healthy controls and after treatment with gabapentin. These changes were not observed after placebo.

Line 539-540: The gabapentin treatments were no better than resting+placebo. Unfortunately your study was not able to show that gabapentin with our without meloxicam is better than rest/placebo in treating NeuP in dogs.

Answer: Please see amended discussion that presents findings in three different perspectives including the caregiver placebo effect (1- treatment vs placebo and caregiver placebo effect with a potential explanation and 2- treatment vs presentation and 3- placebo vs presentation).

Line 577-578: The results of sensory profile (DNIC) are convincing but not the results obtained with the subjective instruments (CSOM, CBPI, CMPS-SF). The fact that results were dependent on the instrument used shows that they are not reliable for the purpose of this study and cannot be used to draw conclusions.

Answer: The conclusion has been rewritten. Thank you very much for your comments. It has clearly improved the manuscript.

---

## [Decision Letter · Decision Letter 1]

11 Nov 2020

Pain burden, sensory profile and inflammatory cytokines of dogs with naturally-occurring neuropathic pain treated with gabapentin alone or with meloxicam

PONE-D-20-22054R1

Dear Dr. Steagall,

We’re pleased to inform you that your manuscript has been judged scientifically suitable for publication and will be formally accepted for publication once it meets all outstanding technical requirements.

Kind regards,

Tamil Selvan Anthonymuthu, Ph. D

Academic Editor

PLOS ONE

Additional Editor Comments (optional):

Reviewers' comments:

Reviewer's Responses to Questions

**Comments to the Author**

1. If the authors have adequately addressed your comments raised in a previous round of review and you feel that this manuscript is now acceptable for publication, you may indicate that here to bypass the “Comments to the Author” section, enter your conflict of interest statement in the “Confidential to Editor” section, and submit your "Accept" recommendation.

Reviewer #1: All comments have been addressed

2. Is the manuscript technically sound, and do the data support the conclusions?

Reviewer #1: Yes

3. Has the statistical analysis been performed appropriately and rigorously? 

Reviewer #1: Yes

4. Have the authors made all data underlying the findings in their manuscript fully available?

Reviewer #1: Yes

5. Is the manuscript presented in an intelligible fashion and written in standard English?

Reviewer #1: Yes

6. Review Comments to the Author

Reviewer #1: Thank you for revising the manuscript. It is much improved and I have no further questions/comments.

7. PLOS authors have the option to publish the peer review history of their article (what does this mean?). If published, this will include your full peer review and any attached files.

Reviewer #1: No

---

## [Editor Report · Acceptance letter]

13 Nov 2020

PONE-D-20-22054R1 

Pain burden, sensory profile and inflammatory cytokines of dogs with naturally-occurring neuropathic pain treated with gabapentin alone or with meloxicam 

Dear Dr. Steagall:

I'm pleased to inform you that your manuscript has been deemed suitable for publication in PLOS ONE. Congratulations! Your manuscript is now with our production department. 

Kind regards, 

on behalf of

Dr. Tamil Selvan Anthonymuthu 

Academic Editor

PLOS ONE